# Enhanced Ion Cluster Size of Sulfonated Poly (Arylene Ether Sulfone) for Proton Exchange Membrane Fuel Cell Application

**DOI:** 10.3390/polym13071111

**Published:** 2021-03-31

**Authors:** Prem P. Sharma, Vo Dinh Cong Tinh, Dukjoon Kim

**Affiliations:** School of Chemical Engineering, Sungkyunkwan University, Suwon 440-746, Gyeonggi, Korea; premsharma15@gmail.com (P.P.S.); vodinhcongtinh@gmail.com (V.D.C.T.)

**Keywords:** PEMFC (proton exchange membrane fuel cell), acid-base, hybrid membrane, proton conductivity), ion cluster, impedance spectroscopy

## Abstract

A successful approach towards enhancement in ion cluster size of sulfonated poly (arylene ether sulfone) (SPAES)-based membranes has been successfully carried out by encapsulating basic pendent branches as side groups. Modified SPAES was synthesized by condensation polymerization followed by bromination with N-bromosuccinamide (NBS) and sulfonation by ring opening reaction. Various molar ratios of branched polyethyleneimine (PEI) were added to the SPAES and the developed polymer was designated as SPAES-x-PEI-y, where *x* denoted the number of sulfonating acid group per polymer chain and y represents the amount of PEI concentration. Polymer synthesis was characterized by ^1^H-NMR (Nuclear magnetic resonance) and FT-IR (Fourier-transform infrared spectroscopy) analysis. A cumulative trend involving enhanced proton conductivity of the membranes with an increase in the molar ratio of PEI has been observed, clearly demonstrating the formation of ionic clusters. SPAES-140-PEI-3 membranes show improved proton conductivity of 0.12 Scm^−1^ at 80 °C. Excellent chemical stability was demonstrated by the polymer with Fenton’s test at 80 °C for 24 h without significant loss in proton conductivity, owing to the suitability of the synthesized hybrid membrane for electrochemical application. Moreover, a single cell degradation test was conducted at 80 °C showing a power density at a 140 mWcm^−2^ value, proving the stable nature of synthesized membranes for proton exchange membrane fuel cell application.

## 1. Introduction

The demand for clean energy has become a main research agenda worldwide as the growth of the population increases with limited resources, and vice-versa. Proton exchange membrane fuel cells (PEMFC), which include a membrane and two electrodes, have grown with huge attraction because of their simple operation and fuel availability [1,2,3,4,5]. In PEMFC, perfluorinated polymer (Nafion) has been commercialized as one polymer backbone which possesses high ionic conductivity with good chemical stability. However, apparent commercialization of this polymer backbone is still hindered because of its high cost and poor performance at a high range of temperature. Thus, the subject of introducing a polymer backbone manufactured with a cheaper material has been the main dilemma for researchers over the last decade [6,7,8,9].

To solve this issue, a variety of polymeric materials have been developed as an alternative for PEMFC, which have operated at a high range of temperature over the last decades [10,11,12]. Arylene main chain polymers such as poly (ether ether sulfone) (PEES), polysulfone (PS), poly (ether ether ketone) (PEEK), and polyphenylene oxides (PPO) are promising candidates next to fluorinated polymers, owing to a characteristic feature of good electrochemical properties [13,14,15]. Most of this study deals with the preparation of membranes by choosing homogeneous ionomers in which the problem of having high water uptake and a swelling ratio with moderate proton conductivities is another issue [16,17,18]. One optional strategy to achieve promising proton conductivity is to lower the area resistance by decreasing the thickness to 50 μm, but mechanical stability would be another problem when employing the membranes into fuel cell application for the long term [19].

An alternate strategy that includes the use of acid-base membrane bases can limit this drawback by controlling the water uptake and swelling ratio with high proton conductivity. Several studies have been reported earlier which include synthesis of acid-base membranes by forming simple physical interactions between a base and stable polymer backbone, blending ionomers with polymer in which the ionomer can form a hydrogen bond, and by forming chemically covalent crosslinked structures [20,21,22,23]. Among the above strategies, covalent crosslink types of membranes possess a drawback of becoming brittle while drying. The physical interaction by hydrogen bonding generates an advantage: the swelling degree could be markedly reduced and the flexible hydrogen interaction by protonated nitrogen in the base could become a leading key factor to limit the brittleness during membrane drying [18,20].

The purpose of the current study is to propagate the cluster size of an acidic polymer by encapsulation of branched pendent basic polymers. The ionic interaction between acid and basic groups present in each form is responsible for enhancing the cluster size, thus we get improved proton conductivity. Further, flexibility to the polymer is supported by hydrogen bond formation. All fundamental parameters and steps involved during polymer backbone synthesis are characterized by ^1^H NMR spectroscopy. The surface morphology is characterized by SEM analysis. PAES-PEI-2 exhibit improves proton conductivity with moderate water uptake and swelling ratio. Further, the single cell degradation test shown by PAES-140-PEI-2 displays an excellent value of power density with respect to the pristine membrane at 80 °C at 100% RH condition, proving the suitability of formed hybrid membranes for PEMFC application.

## 2. Materials and Methods

### 2.1. Materials

Polyethylenimine branced (PEI, Mw = 800, Mn = 600) was obtained from Sigma-Aldrich (St. Louis, MO, USA). 2,2-bis(4-hydroxy-3-methylphenyl)propane (BHMPP), and bis(4-fluorophenyl) sulfone (BFPS)2,2-bis(4-hydroxyphenyl)propane (bis phenol A), N-bromosuccinamide, sodium 3-mercapto-1-propanesulfonate (SMPS), 1,1,2,2-tetrachloroethane (TCE), tetrahydrofuran (THF), benzoyl peroxide (BPO, wetted with ca. 25% water), potassium carbonate anhydrous (K_2_CO_3_), and dimethyl sulfoxide (DMSO) was purchased from TCI (Tokyo, Japan); 2-propanol (CH_3_CHOHCH_3_) and hydrochloric acid (HCl) was obtained from Daejung chemicals ((Seoul, Korea). Nafion solution (5 wt%, EW 1100) was purchased from Dupont ((Wilmington, NC, USA), while platinum (nominally 40% on carbon black, HiSPEC 4000) was purchased from Alfa Aesar (Alpha, NJ, USA).

### 2.2. Synthesis of Methylated Poly(Arylene Ether Sulfone) (PAES-CH_3_)

Condensation polymerization was carried out between monomers having phenol and fluorobenzene groups for synthesizing methylated polyarylene ether sulfone, as shown in Scheme 1. Briefly, BHMPP (7 mmol), bis phenol A (3 mmol) and K_2_CO_3_ (2.4 mmol), were added to a reaction flask containing an azeotropic mixture of DMSO and Toluene (40:60) and equipped with dean stark flask. After complete dissolution, the temperature of the reaction mixture was raised to 150 °C for 4 h. During this step, the formation of water molecules occurred due of the formation of an intermediate complex between K_2_CO_3_ and phenolic derivative. Further, it was subjected to cooling at 50 °C, and an additional amount of BFPS (1 mmol) was added immediately and stirred for 2 h. In a further step, to carry out the complete condensation polymerization, the reaction mixture was subjected for 72 h at 155 °C. The formation of phase-separated grey-colored viscous solution at the bottom of flask indicates the formation of a polymer. Further, it was centrifuged after co-solvent dissolution with THF to remove the salt (KCl) formed during the reaction and was precipitated into 1000 mL of IPA. The obtained white precipitate was then dried at 80 °C under vacuum and designated as PAES-CH_3_.

### 2.3. Synthesis of Bromomethylated Poly(Arylene Ether Sulfone) (PAES-CH_2_Br)

A free-radical substitution reaction with the help of BPO and NBS was carried out to synthesize PAES-CH_2_Br. In detail, 1 mmol of PAES-CH_3_ was dissolved in 150 mL tetrachloroethane in a round bottom flask containing a magnetic stirrer. After complete dissolution, 20 mmol of NBS and 2 mmol of BPO was added to the reaction mixture. The whole reaction was carried out for 24 h at 65 °C. Afterwards the temperature was cooled down to room temperature and precipitated in 1000 mL of IPA. The precipitated product was washed several times with deionized (DI) water to remove impurities and dried in a vacuum oven for 24 h at 80 °C. The obtained product was termed PAES-CH_2_Br.

### 2.4. Synthesis of Sulfonated Poly(Arylene Ether Sulfone) (SPAES)

For sulfonation of bromomethylated forms of poly (arylene ether sulfone) (PAES), SMPS (1.1 mmol) and K_2_CO_3_ were dissolved in 250 mL of DMSO in a round bottom flask equipped with N_2_ and a magnetic bar. Further, the temperature of the solution was increased up to 80 °C for 4 h. Then 0.5 mmol of PAES-CH_2_Br was added to the mixture after cooling the temperature to 40 °C, and this was conducted for 24 h. The orange color of the solution indicates the grafting of the sulfonating group onto the bromomethyl group. This orange color solution was then precipitated into IPA to remove the impurities and further subjected to vacuum dry at 60 °C for 24 h.

### 2.5. Synthesis of Acid-Based Hybrid Membrane

For the synthesis of the hybrid membrane SPAES, poly (arylene ether sulfone) was dissolved into DMSO to make a 10 wt% solution. After the formation of a transparent solution mixture, a predefined amount of PEI i.e., 1, 2, 5 and 10 wt% was added into the solution mixture. The reaction was subjected to vigorous stirring at 100 °C for 24 h. After cooling at room temperature, the solution was casted into a clean glass plate and dried into an oven at 80 °C for 24 h. Furthermore, the formed membrane was dipped into deionized (DI) water for the complete removal of PEI from the membrane. The synthesized membrane was washed several times with DI water and dipped into 1M H_2_SO_4_ for complete ionization of the attached sulfonic acid groups. Synthesized membranes with various PEI concentration are designated as SPAES 140-PEI-1,2,5 and 10, respectively, in Scheme 1.

### 2.6. Characterization

#### 2.6.1. Chemical Structure Analysis

Fourier-Transform Infrared Spectroscopy (FT-IR) spectra were measured using a Perkin-Elmer FT-IR Frontier instrument (Nicolet iS10, Thermo Fisher, Ward Hill, MA, USA). Bromomethylation and sulfonation of polymers was characterized by ^1^H-NMR by 500 MHz Nuclear Magnetic Resonance Spectrometer (^1^H NMR, Varian Unity INOVA 500 MHz, Varian, Palo Alto, CA, USA) by opting DMSO as a solvent. The surface morphology and the size of nanosheets and membranes were evaluated by FE-SEM (EM, Phillip XL30 ESEM-FCG, North Billerica, MA, USA).

#### 2.6.2. Ion Exchange Capacity (IEC)

The IEC of the membranes were calculated by acid-base titration method. The membrane samples were washed with DI water and completely dried to measure their weight (in grams) before being immersed in 1.0 M NaCl solution to complete the exchange of H^+^ ion into Na^+^ ions. The solution was then titrated with 0.1 M of NaOH solution using phenolphthalein as an indicator. The IEC (meq. g^−1^) values of membranes were calculated from Equation (1):(1)$$IEC=\frac{C_{NaOH}\times V_{NaOH\;}}{W_{dry}}$$
where *C_NaOH_* and *V_NaOH_* is the volume of titrated NaOH solution and *W_dry_* is the dry membrane weight.

#### 2.6.3. Proton Conductivity

The membranes were immersed in water and then cut into 3 cm (length) × 1 cm (width) having a thickness of ~80 μm to measure proton conductivity. The sample was placed in the 4-probe cell (BEKKTECH, Loveland, CO, USA) and in-plane proton conductivity was measured by alternating current (AC) impedance spectroscopy (Zahner IM6e, Kronach, Germany) with a frequency range from 1 Hz to 1 MHz at 5 mV under 100% relative humidity. The bulk resistance of the membrane was directly obtained from the impedance curve at different temperatures, and then the proton conductivity of the membrane was calculated using Equation (2):(2)$$\sigma =\;\frac{L}{R\;W\;T}$$
where σ is the proton conductivity of the membrane in (Scm^−1^), L is the distance in the direction of the ion flow between the measurement probes in cm, R is the bulk resistance of the membrane in ohm, W is the width of the membrane in cm, and T is the thickness of the membrane.

#### 2.6.4. Water Uptake and Swelling Ratio

Water uptake is important for the ion exchange membrane because it provides hydrophilicity of the membrane. Transportation of ions takes place mostly through the bound water inside the membrane by a hopping mechanism.

Water uptake was mainly calculated by the following equation
(3)$$\mathrm{WU}\left(\%\right)=\frac{Wet_{w}-Dry_{w}}{Dry_{w}}$$
where *Wet_w_* is the weight of the wet membrane and *Dry_w_* is the weight of the dry membrane.

Hydration number was also calculated by using following equation, where λ is the number of water molecules attached per functional group attached to the polymer backbone, WU is the water uptake, and IEC is the ion exchange capacity (meq. g^−1^).
(4)$$\lambda =\;\frac{10\times \mathrm{WU}}{\mathrm{IEC}\times 18.02}$$

Moreover, the swelling ratio was calculated from Equation (4):(5)$$\mathrm{Swelling}\;\mathrm{ratio}\;=\;\frac{L_{s}{-\;L}_{d}}{L_{d}}$$
where L_s_ is the length of the wet sample and L_d_ is the length of the dry sample, respectively.

#### 2.6.5. Chemical Stability

The SPAES-x-PEI-y sample pieces with a dimension of 3 cm × 1 cm were immersed in 3 M HCl solution for 72 h at room temperature. Each sample was taken out of the solution to be washed with water repeatedly to the physical state. After a regular interval of time, the IC value was measured by the method mentioned earlier. IC values of each membrane were collected to justify its chemical stability.

#### 2.6.6. Oxidative Stability

The anti-oxidation possibility of pristine and hybrid membranes was investigated by measuring the residual weight percentage of each after Fenton’s solution treatment. The completely dry membrane pieces were immersed in the Fenton’s solution (2 wt% H_2_O_2_, 2 ppm FeSO_4_) at 80 °C from 24 h. After completion of the required time, the sample pieces were washed several times with DI water and then dried at 80 °C. The residual weight percentage (RW) was calculated by the difference in the weight of the sample before (m_b_) and after treatment (m_a_) from Equation (6).
(6)$$\mathrm{RW}\left(\%\right)=\frac{m_{a}}{m_{b}}\times 100$$

#### 2.6.7. Membrane Electrode Assembly and Fuel Cell Performance

The catalyst was prepared by taking a mixture of 0.1 g of Pt/C (40%), dispersed into 0.6 g of Nafion ionomer (5 wt% in IPA). Additionally, 1 mL of DI water with 8 g of IPA was added to the mixture of Pt/c and Nafion. It was mixed with the help of a horn-type sonicator (Sonomasher, SL Science, Seoul, Korea) for 30 min. This sonication was repeated two times with an interval of 15 min. The mixture was sprayed onto carbon paper to prepare a gas diffusion layer (GDL). The membrane electrodes assembly (MEA) was prepared by pressing the catalyst-coated membrane using a heating press (Ocean Science, Seoul, Korea) at 110 °C and 5 MPa for 3 min. The active area of the MEA for this process was 6.25 cm^2^ and Pt loading amount for both anode and cathode was 0.6 mg cm^−2^ each. The fuel cell performance was measured using a unit cell station (SPPSN-300) provided by CNL Energy (Seoul, Korea). During the cell test, hydrogen and oxygen gas was continuously fed to anode and cathode sites at the flow rate of 300 cm^3^ per min, respectively. The fuel cell performance was measured at 80 °C under 100% relative humidity (RH).

## 3. Results and Discussion

### 3.1. Chemical Structure Characterization

Chemical structure of the synthesized polymer was revealed by ^1^H-NMR spectroscopy analysis (Figure 1a–c). In this first step, the methylated groups, which are attached to the main polymer backbone of PAES, were indicated by two distinct peaks appearing at the chemical shift value of 1.7 and 2.06 ppm [23]. Another broad peak in the region 6–8 ppm was due to the aromatic region of PAES structure (Figure 1a) [24]. Further, in the second step, the peak arose at 4.5 ppm, indicating the attachment of -CH_2_Br group to the PAES backbone. The occurrence of this peak at this chemical shift value also confirms the conversion of methylated group into the bromomethylated group (Figure 1b). The third step was covered by the grafting of the sulfonated group present in SMPS to the PAES polymer backbone through a ring-opening reaction. The appearance of new peaks at a chemical shift value of 2.50 and 2.40 ppm could be predicted by the protons attaching the -CH_2_- group with the sulfide linkage and -SO_3_H- of SMPS, respectively **(**Figure 1c). Another confirmatory analysis was carried out by FT-IR analysis with the occurrence of a peak at 1050 cm^−1^, indicating the presence of a sulfonating group [25].

The surface morphology of the synthesized membrane was characterized by FE-SEM analysis (Figure 2). The rate of leaching of PEI from the polymer matrix plays an important role in forming microphase separation. As we increase the PEI content, the rate of solubility will be faster, which corresponds to the solvent exchange rate of water with that of PEI. Moreover, increase in the molar ratio of PEI due to the solubility parameter difference between SPAES and PEI with polar solvent results in the increment of microphase separated morphology (Figure 2c,d). Microphase separation may produce micropores on the surface of the polymer leading to the increase in the ion cluster size, due to which the conduction of the proton will be on the higher side. Secondly, phase separation morphology which occurred in the membrane might be due to the different nature of hydrophobic PAES and hydrophilic PEI, which may be varied by the respective molar ratios. At this time, the polymer chain may provide more free space for ease in conduction of protons and could prevent aggregation of sulfonic acid groups.

### 3.2. Water Uptake and Swelling Ratio

Water uptake and swelling ratio are another important phenomenon for a membrane to be used in the fuel cell application, as seen from Figure 3a,b. The water uptake value is increased with increasing PEI content. The same trend was shown by hydration number, as it is the number of water molecules absorbed in the functional group. This was found to be a maximum ~58% in SPAES-140-PEI-4. This is because of the formation of the more ionic cluster, which also deteriorates the flexibility of the membrane. It was further justified by checking the swelling degree which demonstrates the same behavior as SPAES-140-PEI-4, and the value is 32.65%, which is approximately two-fold higher than that of SPAES and 1.05% higher that of SPAES-140-PEI-3. High water uptake and swelling ration could cause deterioration in the mechanical stability of the membrane for application in electrolyzers [26].

### 3.3. Proton Conductivity

Proton conductivity with the respective hybrid membrane has been shown in Figure 4, with different temperatures ranging under the 100% relative humidity condition. As we can see, the proton conductivity gets increased as we increase the PEI content, and the maximum value was shown by the membrane PAES-140-PEI-3, which is 0.124 Scm^−1^ and the corresponding value of water uptake is 44%. Thus, it can be stated that the mobility of the ions with water gets enhanced because water molecules can act as additional proton carriers by vehicular mechanism. This can be done by the rapid exchange of protons by hydrogen bridges. Moreover, ionic clusters formed by the branched PEI in the polymer backbone could lead to the formation of some ionic channels for facilitating the proton transport by hopping and vehicular mechanisms. On the other hand, the higher basicity of PEI could also be another factor for enhanced proton conductivity. Because high basicity can make an easier pathway for proton conduction by the hopping mechanism and the surface hopping transport, PEI content may also be considered as a proton transport mechanism that becomes important by increasing the concentration of PEI. The protonation-deprotonation loop formation between two different sites is responsible for strengthening the proton conductivity by lowering the ion transfer barrier [27].

### 3.4. Chemical Stability and Oxidative Stability

Chemical stability and oxidative stability are the important parameters for the membrane because the durability of the membrane is mainly decided by these parameters. This was checked by dipping the samples of membranes into Fenton’s reagent (2 wt% H_2_O_2_ and 2 ppm FeSO_4_) at 80 °C for 24 h. The residual weight was found to be 74.45% for pristine and 72.65% and 58.23% for SPAES-140-PEI-3 and 4, respectively. The current study has the pendent side (sulfonate propyl groups) grafted into the PAES structure by the sulfide linkages, which can change form to sulfone groups under the attaching of •OH and •OOH radicals. Secondly, being a long carbon chain, it tends to keep the radicals away from the polymer backbone, and due to this, the degradation of polymer is on slower side, as seen in the Fenton’s test. Moreover, it can be observed that the membrane containing PEI was able to uptake a large amount of water because of the formation of a lot of ionic clusters. Thus, the •OH and •OOH radicals easy attached on the membrane to degrade the polymer structure at the ether bonding [28,29,30].

Further, one more parameter was also checked to predict the acidic stability of the synthesized hybrid membrane. For this, the samples of the membrane were dipped into 3 M HCl solution for 72 h and conductivity was checked over a periodic interval of 12 h. A minimal change less than 1% was found for all synthesized membranes shown in Figure 5. The long flexible chain was responsible for providing protection to the polymer backbone.

## 4. Cell Performance

The single-cell fuel cell test was performed at 80 °C under a 100% relative humidity condition for SPAES-140-PEI-3. Open circuit voltage value for the membrane was 0.96 V and the maximum power density of 144 mWcm^−2^ was achieved by the hybrid membrane (Figure 6). The sulfide linkage in the polymer backbone provided stability to the polymer backbone; further, the sulfonating group attached with the long flexible chain provided steric hindrance to polymers from the attack of radicals forming during the cell test. Due to this reason, hybrid membranes displayed a very slow rate of voltage drop with increasing current density value, proving the durability of the membrane for fuel cell application. On the other hand, the transfer of the proton through the membrane was very well facilitated by the formed ionic cluster from basic pendant group.

## 5. Conclusions

In the current study, a series of acid-based hybrid membranes containing SPAES as a polymer backbone were successfully synthesized and characterized. A detailed study involving in the encapsulation of PEI by forming ionic interactions was responsible for enhancing the ionic cluster size and was analyzed systematically on the basis of chemical structure analysis by ^1^H-NMR, FTIR and SEM analysis, respectively. An enhancement in the proton conductivity value was observed with the optimum water uptake and swelling ratio. A maximum value of 0.12 Scm^−1^ was successfully achieved by SPAES-140-PEI-3 at 80 °C, proving the enhanced ionic cluster size. Additionally, to check the performance of the best membrane, it was further subjected to an H_2_/O_2_ single fuel cell test at 80 °C, showing the maximum power density value of 140 mWcm^−2^ and proving its potential as a promising candidate for electrochemical applications.

Further, more optimization and improvement towards controlling the water uptake, swelling ratio and enhanced chemical stability with higher proton conductivity is ongoing as a future work to synthesize durable membranes for fuel cell application.

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
