# Peer review of "Enhanced Ion Cluster Size of Sulfonated Poly (Arylene Ether Sulfone) for Proton Exchange Membrane Fuel Cell Application"

_polymers, 2021, doi:10.3390/polym13071111_

Round 1

Reviewer 1 Report

This manuscript entitled “Enhanced Ion Cluster Size of Sulfonated Poly(Arylene Ether Sulfone) for Proton Exchange Membrane Fuel Cell Application” well fit into the journal’s scope. In this study, various molar ratios of branched polyethyleneimine (PEI) were added to the SPAES to developed SPAES-x-PEI-y. Synthesized membranes were characterized by and 1H-NMR and FT-IR, as well as the protonic conductivity and single-cell degradation test, were conducted. But still, this manuscript requires a major revision for getting published.

Here are a few comments for improvements,

  • No references have been cited beyond the introduction of the manuscript. In results and discussion, many evidence-based on literature for the present study was given, those statements should be cited with appropriate references.
  • In the abstract, “..show improved ionic conductivity of 0.12 Scm-1 at 80 oC” it should be protonic conductivity. Use the same terms throughout the manuscript.
  • It is mentioned that the protonic conductivity was directly obtained from the impedance curve (bulk resistance & hydroxide ion conductivity of the membrane) by using them in equation 2. So the impedance curve for any one of the samples as an example for the statement. As well as, provide the numerical values of the other parameters in equation 2 (such as L, W, T) that are required to determine the conductivity.
  • The following statement requires an additional explanation. “Because high basicity can make easier pathway for proton conduction by hopping mechanism and the surface hopping transport with PEI content may also be considered as proton transport mechanism that becomes important by increasing the concentration of PEI.”
  • It is mentioned, “Fig.3(b) is elucidating the possible mechanism of transport of proton.” Provide figure 3(b), and explain the mechanism in detail.
  • Findings from session 3.3. (water uptake) was used to justify the results in session 3.2. (protonic conductivity). In that case, suggested rearranging the order of the sessions, present session 3.3. before 3.2.
  • In Figure 6, the Polarization curve for SPAES-140-PEI-3 was shown. But the Figure 5 shows that the SPAES-140-PEI-4 has superior protonic conductivity in acidic conditions. Hope based on the protonic conductivity in basic condition SPAES-140-PEI-3 was selected for the single-cell degradation test. Added as justification in session 4 for choosing SPAES-140-PEI-3.
  • The manuscript should be checked for typo errors. Fig 3(b) and Table 1 are missing.

Author Response

This manuscript entitled “Enhanced Ion Cluster Size of Sulfonated Poly (Arylene Ether Sulfone) for Proton Exchange Membrane Fuel Cell Application” well fit into the journal’s scope. In this study, various molar ratios of branched polyethyleneimine (PEI) were added to the SPAES to developed SPAES-x-PEI-y. Synthesized membranes were characterized by and 1H-NMR and FT-IR, as well as the protonic conductivity and single-cell degradation test, were conducted. But still, this manuscript requires a major revision for getting published.

Here are a few comments for improvements,

  • No references have been cited beyond the introduction of the manuscript. In results and discussion, many evidence-based on literature for the present study was given, those statements should be cited with appropriate references.

Ans: Thank you for the suggestion. The appropriate references have been cited in the revised manuscript. (Pag.10-14)

  • In the abstract, “show improved ionic conductivity of 0.12 Scm-1 at 80 oC” it should be protonic conductivity. Use the same terms throughout the manuscript.

Ans: Thanks for the suggestion. It has been replaced throughout the manuscript. (Pag.1 and Pag.8).

  • It is mentioned that the protonic conductivity was directly obtained from the impedance curve (bulk resistance & hydroxide ion conductivity of the membrane) by using them in equation 2. So, the impedance curve for any one of the samples as an example for the statement. As well as, provide the numerical values of the other parameters in equation 2 (such as L, W, T) that are required to determine the conductivity.

Ans: Thanks for the reviewer’s kind comment. As mentioned in the manuscript, the bulk resistance of membrane was directly obtained from the impedance curves at different temperature (Not the protonic conductivity as reviewer comment). And then the proton conductivity was calculated using the equation 2. Moreover, the number values of thickness (T, ~80 cm), length (L, ~3 cm) and width (W, ~1 cm) were mentioned in the main text. (page 8).

On the other hand, the authors realized that the hydroxide ion conductivity of membrane are not relative the proton conductivity of membrane. Thus, it was removed out of the manuscript.

The following statement requires an additional explanation. “Because high basicity can make easier pathway for proton conduction by hopping mechanism and the surface hopping transport with PEI content may also be considered as proton transport mechanism that becomes important by increasing the concentration of PEI.”

Ans: An additional explanation involving chemistry is added to the revised manuscript (Pag.14, Paragraph.1).

  • It is mentioned, “Fig.3(b) is elucidating the possible mechanism of transport of proton.” Provide figure 3(b), and explain the mechanism in detail.

Ans: Thank you for the suggestion. It was inserted by mistake and following typo error has been removed in the revised manuscript.

  • Findings from session 3.3. (water uptake) was used to justify the results in session 3.2. (protonic conductivity). In that case, suggested rearranging the order of the sessions, present session 3.3. before 3.2.

Ans: Thank you for the suggestion. Following section has been rearranged in the revised manuscript (Pag.13-14).

  • In Figure 6, the Polarization curve for SPAES-140-PEI-3 was shown. But the Figure 5 shows that the SPAES-140-PEI-4 has superior protonic conductivity in acidic conditions. Hope based on the protonic conductivity in basic condition SPAES-140-PEI-3 was selected for the single-cell degradation test. Added as justification in session 4 for choosing SPAES-140-PEI-3.

Ans: Thank you for the comment. SPAES-140-PEI-3, was chosen for single-cell degradation test by investigating the water uptake and swelling ratio respectively with the different synthesized membranes. On the other hand, SPAES-140-PEI-4 has superior protonic conductivity in acidic conditions but due to high water uptake at 800C and in 100 % RH condition it could degraded rapidly.

  • The manuscript should be checked for typo errors. Fig 3(b) and Table 1 are missing.

Ans: It has been checked and corrected in the revised manuscript.

Reviewer 2 Report

Demand of clean energy has become the main research agenda worldwide as growth of population increases with limited resources and vice-versa. Proton exchange  membrane fuel cell (PEMFC) which includes a membrane, and two electrodes, has grown up with huge attraction because of its simple operation and fuel availability. The proton exchange membrane fuel cells (PEMFCs) are promising energy devices for stationary and mobile applications because of high power density, high efficiency, low operating temperature, low emissions, low noise, and great environmental compatibility. The PEMFCs are composed of gas diffusion layer (GDL) including gas diffusion backing (GDB) and microporous layer (MPL), membrane electrode assembly (MEA), and bipolar plates with gas channels. The fibrous gas diffusion layer is a core component of a PEMFC, enabling transport of gases, liquids and electricity within the cell. In this paper, modified (SPAES) was synthesized by condensation polymerization followed by bromination with N-bromosuccinamide (NBS) and sulfonation by ring opening reaction. Various molar ratios of branched polyethyleneimine (PEI) were added to the SPAES and the developed polymer was designated as SPAES-x-PEI-y, where x denoted the number of sulfonating acid group per polymer chain and y represents the amount of PEI concentration. The topic may be important, the results are interesting and the methodology followed is appropriate, while the content falls well within the scope of this Journal. In general the paper makes fair impression and my recommendation is that it merits publication in this Journal, after the following major revision:

  1. The current one is nothing but a literature review. Why their work is important comparing to previous reports? I think this is essential to keep the interest of the reader.
  2. In Fig.3, 4(a) and 5, the authors should give the explanations for the difference of data collected from different sources.
  3. Materials and Methods part. Although the results look “making sense”, the current form reads like a simple lab report. The authors should dig deeper in the results by presenting some in-depth discussion.
  4. A cumulative trend involving enhanced proton conductivity of the membranes with increasing in the molar ratio of PEI has been observed clearly demonstrated the formation of ionic clusters. SPAES-140-PEI-3, membranes show improved ionic conductivity of 0.12 Scm-1 at 800C. Excellent chemical stability was demonstrated by the polymer with Fenton’s test at 800C for 24 hours without significant loss in proton conductivity owing the well suitability of synthesized hybrid membrane for electrochemical application. Moreover, single cell degradation test was conducted at 800C showing a power density 140 mWcm-2 value proving the stable nature of synthesized membrane for proton exchange membrane for fuel cell application. The authors should give some explanation on above results and data.
  5. Proton exchange membrane fuel cells have attracted attention from energy devices such as portable, mobile and stationary devices, since it helps effective reductions of energy shortage and environment pollution. The present work mainly focuses on lab work. It does not necessarily imply that the theoretic work (modeling) is not important. The authors omit this part during the current literature review, which should include a brief review of the theoretic work after revision. In the theoretic perspective, fractal theory is a very important tool, which can be used to investigate proton exchange membrane fuel cells, see [International Journal of Hydrogen Energy, 2018, 43(37):17880-17888; Fractals, 2019, 27(2):1950012]. Authors should introduce some related knowledge to readers.
  6. Please, expand the conclusions in relation to the specific goals and the future work.

Author Response

Demand of clean energy has become the main research agenda worldwide as growth of population increases with limited resources and vice-versa. Proton exchange membrane fuel cell (PEMFC) which includes a membrane, and two electrodes, has grown up with huge attraction because of its simple operation and fuel availability. The proton exchange membrane fuel cells (PEMFCs) are promising energy devices for stationary and mobile applications because of high power density, high efficiency, low operating temperature, low emissions, low noise, and great environmental compatibility. The PEMFCs are composed of gas diffusion layer (GDL) including gas diffusion backing (GDB) and microporous layer (MPL), membrane electrode assembly (MEA), and bipolar plates with gas channels. The fibrous gas diffusion layer is a core component of a PEMFC, enabling transport of gases, liquids and electricity within the cell. In this paper, modified (SPAES) was synthesized by condensation polymerization followed by bromination with N-bromosuccinamide (NBS) and sulfonation by ring opening reaction. Various molar ratios of branched polyethyleneimine (PEI) were added to the SPAES and the developed polymer was designated as SPAES-x-PEI-y, where denoted the number of sulfonating acid group per polymer chain and y represents the amount of PEI concentration. The topic may be important, the results are interesting and the methodology followed is appropriate, while the content falls well within the scope of this Journal. In general, the paper makes fair impression and my recommendation is that it merits publication in this Journal, after the following major revision:

  1. The current one is nothing but a literature review. Why their work is important comparing to previous reports? I think this is essential to keep the interest of the reader.

Ans. Thanks for the reviewer’s kind comment. The degradation of the pendent groups is very important in fuel cell application because it is directly related with the degradation of performance during their application. Thus, in this study, the pendent side (sulfonate propyl groups) was grafted into PAES structure by the sulfile linkages which can change form to sulfone groups under attaching of •OH and •OOH radicals.

[1] Viviani, Marco, et al. "Highly stable membranes of poly (phenylene sulfide benzimidazole) cross-linked with polyhedral oligomeric silsesquioxanes for high-temperature proton transport." ACS Applied Energy Materials 3.8 (2020): 7873-7884.

[2] D. Zhao, J. Li, M.K. Song, B. Yi, H. Zhang, M. Liu, A Durable Alternative for Proton‐Exchange Membranes: Sulfonated Poly (Benzoxazole Thioether Sulfone) s, Adv. Energy Mater. 1 (2011) 203-211.

Moreover, the authors want to improve the proton conductivity of SPAES membrane by using different contents of PEI. It can orientate sulfonic acid groups of SPAES polymer to form ionic cluster. Thus, a huge amount of the ionic clusters are formed with the presence of PEI in the membrane.

  1. In Fig.3, 4(a) and 5, the authors should give the explanations for the difference of data collected from different sources.

Ans. Thanks for the reviewer’s suggestion. The discussion of the Fig. 3, 4(a) and 5 were rewritten in the main text. Moreover, the equations related to date collection is represented as equation-2, 3 and 5 respectively. (Page. 8-9).

  1. Materials and Methods part. Although the results look “making sense”, the current form reads like a simple lab report. The authors should dig deeper in the results by presenting some in-depth discussion.

Ans: Thank you for the kind comment. To make it more scientific the references related to the structure and performance of the polymer has been added to the result and discussion part. (Pag.10-14).

  1. A cumulative trend involving enhanced proton conductivity of the membranes with increasing in the molar ratio of PEI has been observed clearly demonstrated the formation of ionic clusters. SPAES-140-PEI-3, membranes show improved ionic conductivity of 0.12 Scm-1 at 800C. Excellent chemical stability was demonstrated by the polymer with Fenton’s test at 800C for 24 hours without significant loss in proton conductivity owing the well suitability of synthesized hybrid membrane for electrochemical application. Moreover, single cell degradation test was conducted at 800C showing a power density 140 mWcm-2 value proving the stable nature of synthesized membrane for proton exchange membrane for fuel cell application. The authors should give some explanation on above results and data.

Ans: Thank you for the comment. The sufide linkage present in the polymer as a side chain is quite stable and has a tendency to change in to sulfone group due to oxidation when counter attacked by radicals generated during oxidative environment. Moreover, being a long carbon chain, it tends to keep the radicals away from the polymer backbone and due to which the degradation of polymer is on slower side as seen in the Fenton’s test. Thus, when it is applied for single cell test the voltage drop is also slow. It is rewritten and added in the revised manuscript. (Pag.15, Paragraph.1).

  1. Proton exchange membrane fuel cells have attracted attention from energy devices such as portable, mobile and stationary devices, since it helps effective reductions of energy shortage and environment pollution. The present work mainly focuses on lab work. It does not necessarily imply that the theoretic work (modeling) is not important. The authors omit this part during the current literature review, which should include a brief review of the theoretic work after revision. In the theoretic perspective, fractal theory is a very important tool, which can be used to investigate proton exchange membrane fuel cells, see [International Journal of Hydrogen Energy, 2018, 43(37):17880-17888; Fractals, 2019, 27(2):1950012]. Authors should introduce some related knowledge to readers.

Ans: Thank you for such a nice recommendation. It will be really beneficial for the readers to get in touch with such excellent research work. As per suggestion it has been added to the introduction section and cited as well.

  1. Please, expand the conclusions in relation to the specific goals and the future work.

Ans: The conclusion part has been expanded and rewritten again including future works. (Pag.17-18).
